# OVERCOMING DATA INEQUALITY ACROSS DOMAINS WITH SEMI-SUPERVISED DOMAIN GENERALIZATION

## ABSTRACT

While there have been considerable advancements in machine learning driven by extensive datasets, a significant disparity still persists in the availability of data across various sources and populations. This inequality across domains poses challenges in modeling for those with limited data, which can lead to profound practical and ethical concerns. In this paper, we address a representative case of data inequality problem across domains termed Semi-Supervised Domain Generalization (SSDG), in which only one domain is labeled while the rest are unlabeled. We propose a novel algorithm, ProUD, designed for progressive generalization across domains by leveraging domain-aware prototypes and uncertainty-adaptive mixing strategies. Our experiments on three different benchmark datasets demonstrate the effectiveness of ProUD, outperforming existing baseline models in domain generalization and semi-supervised learning.

## 1 INTRODUCTION

In the realm of machine learning, the availability of extensive datasets has played a pivotal role in driving advancements (Sun et al., 2017; Kaplan et al., 2020). However, acquiring sufficient training data remains a challenge due to data accessibility disparities across different sources and populations, an issue commonly termed as *data inequality*. The World Development Report by the World Bank (Bank, 2021) underlines this problem, noting that developing economies frequently grapple with data scarcity stemming from an absence of requisite infrastructure for data connectivity, storage, and processing. The deficiency also extends to the limited availability of human expertise and skilled labor in these areas.

Such data inequality not only presents practical challenges but also raises ethical concerns in the design and deployment of machine learning models. This issue is prevalent across various fields, with a clear example in the biomedical sector. Data inequality across ethnic groups can lead to uneven model performance, thereby exacerbating global healthcare inequalities, as evidenced by Gao & Cui (2020); Gao et al. (2023). In particular, recent statistics (Guerrero et al., 2018) reveal a severe imbalance: data from 416 cancer-related genome-wide association studies were collected from Caucasians (91.1%), followed distantly by Asians (5.6%), African Americans (1.7%), Hispanics (0.5%), and other populations (0.5%). It is noteworthy that non-Caucasians, constituting approximately 84% of the world's population, find themselves at a considerable data disadvantage. This inequality can lead machine learning models to exhibit compromised predictive accuracy and a lack of robustness for these underrepresented groups, potentially harming their healthcare outcomes (Martin et al., 2019; Rajkomar et al., 2018).

In light of these concerns, we address a representative case of the data inequality problem across domains, termed Semi-Supervised Domain Generalization (SSDG). More specifically, the core objective of SSDG is to learn domain-invariant features from multiple source domains, wherein only one domain is labeled while the rest domains remain unlabeled. Such a setting mirrors real-world situations, especially when obtaining labeled data from certain domains is considerably more difficult than from others. Table 1 provides some examples of this scenario across various fields.

To mitigate the issue of data inequality across domains with SSDG, we propose a Prototype-based Uncertainty-adaptive Domain Generalization algorithm, denoted as ProUD. The main challenge in addressing data inequality across domains stems from the need to efficiently leverage the potential of samples from unlabeled source domains. Consequently, within our ProUD method, we integrate

Table 1: Examples of machine learning applications in various fields susceptible to data inequality problem across domains, where Semi-Supervised Domain Generalization(SSDG) can be applied.

| Task / Application | Labeled Domain | Unlabeled Domain |
|---|---|---|
| Biomedical Imaging | Caucasians | Other ethnicities |
| | Central hospitals | Peripheral hospitals |
| Natural Language Processing | English | Minority languages |
| | Standard language | Regional dialects |
| Autonomous Driving | Urban area | Rural area |
| | Typical weather conditions | Rare weather conditions |
| Agricultural Crop Monitoring (Using Satellite Images) | Developed countries | Developing countries |
| | Commercial satellites | Non-commercial satellites |

prototype-based mechanisms designed to assign reliable pseudo-labels to such unlabeled data and learn feature representations invariant across different domains. These include (1) prototype-based pseudo-labeling (ProtoPL), (2) uncertainty-adaptive integration of unlabeled domains (DomainMix), and (3) contrastive learning for domain-invariant representations (Prototype Merging Loss).

We compare our method with extensive baselines including domain generalization (DG) and semi-supervised learning (SSL) methods. However, both DG and SSL approaches exhibit suboptimal performance when applied to the data inequality problem with SSDG. The limitation stems from DG methods not being designed to utilize unlabeled data and SSL methods assuming that all training data are sampled from a single distribution. Thus, we include stronger baselines by incorporating domain adaptation (DA) and domain generalization (DG), employing a direct approach to SSDG consisting of two training phases: pseudo-labeling and domain generalization. Furthermore, we also compare our method with EID (Lin et al., 2023), a straightforward approach to address SSDG by generating pseudo-labels for unlabeled data and filtering out noisy labels. Extensive experiments have been conducted to compare our approach with these baselines, illustrating superior overall performance and robustness on three different datasets.

Overall, our contributions can be summarized as follows:

- We introduce a novel algorithm, $\mathrm{ProUD}$, to address SSDG, a representative case of data inequality, where only one source domain is labeled while the others are unlabeled.

- We propose a prototype-based pseudo-labeling method (ProtoPL) that utilizes domain-aware prototypes to assign reliable pseudo-labels to the samples from a set of multiple unlabeled domains.

- We present a progressive generalization method (DomainMix) that enables gradual integration of unlabeled domains by utilizing entropy-based uncertainty.

- We develop a contrastive loss (Prototype Merging Loss) for learning domain-invariant features by merging domain-aware prototypes.

- Our method consistently outperforms extensive baseline models, demonstrating superior average performance and robustness across various domain combinations on three different datasets.

## 2 RELATED WORK

**Domain Generalization**  Domain Generalization (DG) aims to train a model using data from one or multiple source domains, enabling it to achieve effective generalization across unseen target domains. Early research on DG primarily focused on acquiring domain-invariant representations by aligning features across distinct sources (Gan et al., 2016; Ghifary et al., 2016; 2015; Li et al., 2018b). Meta-Learning has also been explored for DG (Balaji et al., 2018; Dou et al., 2019; Li et al., 2018a; 2019), involving training a model with pseudo-train and pseudo-test domains drawn from source domains to simulate domain shifts. Another approach to DG incorporates data augmentation, which seeks to increase style diversity at the image-level (Shankar et al., 2018; Zhou et al., 2020a; Xu et al., 2021; Gong et al., 2019; Zhou et al., 2020b), or the feature-level (Zhou et al., 2021a; Li et al., 2022; Zhong et al., 2022; Li et al., 2021). However, these DG approaches are limited by their heavy reliance on accessing labeled data from multiple source domains. To address

practical scenarios, single-source DG methods (Romera et al., 2018; Volpi et al., 2018; Zhao et al., 2020) have emerged, which utilize labeled data from a single domain. Single-source DG methods are largely built upon adversarial domain augmentation (Huang et al., 2020; Qiao et al., 2020; Wang et al., 2021). Our problem setting, SSDG, is similar to single-source DG, as it utilizes only a single labeled source domain during the training process. However, it differs in that it also has access to unlabeled data from multiple domains. Our work focuses on effectively leveraging the unlabeled source data by progressively mixing it with labeled source data through DomainMix.

**Semi-Supervised Learning**   In semi-supervised learning (SSL), a small amount of labeled data is accessible alongside a larger volume of unlabeled data to train a model. Various methods have been developed to effectively utilize this mix of labeled and unlabeled data. An established approach in SSL is consistency regularization, which forces a model's predictions to remain consistent when alterations are made to the model (Laine & Aila, 2016; Tarvainen & Valpola, 2017; Miyato et al., 2018). Another popular approach is to employ pseudo-labeling (Lee et al., 2013), which generates pseudo-labels for unlabeled data using a pretrained model (Xie et al., 2020). Furthermore, MixMatch (Berthelot et al., 2019), FeatMatch (Kuo et al., 2020), and FixMatch (Sohn et al., 2020) incorporate a combination of pseudo-labeling and consistency regularization. However, typical SSL methods heavily rely on the assumption that labeled and unlabeled data share an identical distribution, which can be quite challenging to fulfill in real-world scenarios. SSDG can be viewed as a particular variation of SSL, specifically relevant in practical situations where the labeled and unlabeled data hold different distributions. The difference in distribution between the labeled and unlabeled data can result in significant bias in the pseudo-labels, leading to degradation in performance. To address this issue, EID (Lin et al., 2023) filters out noisy labels using a specified cleaning rate to generate a set of pseudo-labeled data with enhanced quality, employing a dual network architecture. However, this approach can be inefficient in terms of harnessing the potential of unlabeled data, as it only utilizes a fraction of the available data during the training process. Rather than applying a threshold to filter out noisy labels, our approach involves a measurement of the uncertainty associated with each pseudo-label, to effectively utilize them during the training process.

**Mixup**   Mixup (Zhang et al., 2017) is a simple yet effective method to extend the training data distribution, founded on the intuition that performing linear interpolations among input images will result in corresponding linear interpolations of the labels. As an intuitive strategy for data augmentation, Mixup has also been studied in the context of DG (Zhou et al., 2021a; Wang et al., 2020; Xu et al., 2021; Lu et al., 2022). FIXED (Lu et al., 2022) states two limitations of applying Mixup to DG: first, in discerning domain and class information, which leads to performance degradation due to entangled domain-class knowledge. Second, Mixup may generate noisy data, especially when data points from distinct classes are close to each other. Inspired by this research, we exclusively apply DomainMix to images with the same classes before Mixup, preventing the entanglement of domain-class knowledge. Additionally, we utilize Prototype Merging Loss to ensure that data points from different classes remain well-separated. We employ DomainMix at the feature-level for effective generalization, as demonstrated by Zou et al. (2023); Li et al. (2021); Verma et al. (2019) as a more promising direction for DG.

## 3   METHODOLOGY

In this section, we introduce the problem setting for Semi-Supervised Domain Generalization (SSDG) and present our proposed method, termed the Prototype-based Uncertainty-adaptive Domain Generalization (ProUD) algorithm.

### 3.1   SEMI-SUPERVISED DOMAIN GENERALIZATION

In this work, we tackle the $K$-way image classification problem for Semi-Supervised Domain Generalization (SSDG). The image space $\mathcal{X}$ and label space $\mathcal{Y}$ are assumed to be shared across all domains. For notational simplicity, the notation $\mathcal{D}_t$ denotes both the $t$-th domain and the dataset sampled from it interchangeably. To formulate the SSDG problem, consider a training dataset $\mathcal{D}^{\text{train}} = \mathcal{D}^l \cup \mathcal{D}^u$, consisting of a single labeled source domain $\mathcal{D}^l$ and a set of unlabeled source domains $\mathcal{D}^u$. Specifically, we have a labeled source domain $\mathcal{D}^l = \mathcal{D}_0 = \{(\boldsymbol{x}_0^{(i)}, y_0^{(i)})\}_{i=1}^{N_0}$, where $N_0$ denotes the total number of samples in $\mathcal{D}_0$, and $\boldsymbol{x}_0^{(i)}$ and $y_0^{(i)}$ indicate input data and its corresponding label of the

---

**Algorithm 1** Prototype-based Uncertainty-adaptive Domain Generalization (ProUD)

---

**Input:** Pretrained model $f = h \circ g$, labeled source domain dataset $\mathcal{D}^l = \mathcal{D}_0$, unlabeled source domain datasets $\mathcal{D}^u = \{\mathcal{D}_1, \ldots, \mathcal{D}_T\}$, and balancing parameter $\alpha$.
**for** epoch $= 1$ to $E$ **do**                                                                          ▷ for each epoch
    **for** $t = 1$ to $T$ **do**                                                          ▷ for each unlabeled source domain
        Apply ProtoPL on $\mathcal{D}_t$ to generate domain-aware prototypes $\{C_{t,k}\}_{k=1}^K$.
        Based on them, build a pseudo-labeled dataset $\widetilde{\mathcal{D}}_t$ with uncertainty estimates.
    **end for**
    Sample a sequence of mini-batches $\{\mathcal{B}_s^u\}_{s=1}^S$ from $\bigcup_{t=1}^T \widetilde{\mathcal{D}}_t$
    **for** $s = 1$ to $S$ **do**                                                          ▷ for each mini-batch
        $\mathcal{B}_s^l = \text{SampleMatch}(\mathcal{B}_s^u, \mathcal{D}_0)$
        $\mathcal{B}_s^m = \text{DomainMix}(\mathcal{B}_s^l, \mathcal{B}_s^u)$
        Update the parameters of $f$ with $\mathcal{L}(h \circ g) = \mathcal{L}_{\text{CE}}(h \circ g; \mathcal{B}_s^m) + \alpha \mathcal{L}_{\text{PML}}(g; \mathcal{B}_s^l \cup \mathcal{B}_s^u)$
    **end for**
**end for**

---

$i$-th sample. On the other hand, a set of unlabeled source domains $\mathcal{D}^u = \{\mathcal{D}_t\}_{t=1}^T$ does not contain any label information (*i.e.*, $\mathcal{D}_t = \{\boldsymbol{x}_t^{(i)}\}_{i=1}^{N_t}$). In this setting, we have a model $f(\boldsymbol{x}) = (h \circ g)(\boldsymbol{x})$, where $g : \mathbb{X} \to \mathbb{R}^d$ and $h : \mathbb{R}^d \to \mathbb{R}$ represent the feature extractor and the classifier, respectively. $d$ denotes the feature dimension. The goal of SSDG is to train the model $f$ with $\mathcal{D}^{\text{train}}$ to generalize well on a test dataset from unseen domain $\mathcal{D}^{\text{test}}$, where $\mathcal{D}^{\text{train}} \cap \mathcal{D}^{\text{test}} = \emptyset$.

## 3.2 THE PROUD ALGORITHM

To address the challenge of SSDG, we present the Prototype-based Uncertainty-adaptive Domain Generalization (ProUD) algorithm, which is detailed in Algorithm 1. At a high level, ProUD is characterized by its utilization of (1) prototype-based pseudo labeling (ProtoPL), (2) uncertainty-adaptive integration of unlabeled domains (DomainMix), and (3) contrastive learning for domain-invariant representations (Prototype Merging Loss). Given labeled source domain $\mathcal{D}^l = \mathcal{D}_0$ and unlabeled source domains $\mathcal{D}^u = \{\mathcal{D}_1, \ldots, \mathcal{D}_T\}$, we first pretrain the model $f$ using $\mathcal{D}^l$. Following this, for the main training, we use ProtoPL to generate domain-aware prototypes and pseudo-labels with uncertainty estimates for $\mathcal{D}^u$ in the beginning of every epoch. For each mini-batch, we employ DomainMix to progressively achieve generalization through the gradual integration of $\mathcal{D}^u$ based on uncertainty. Finally, the model is optimized using cross entropy and Prototype Merging Loss, aiming to achieve discrimination between classes and generalization across domains simultaneously.

**Pretraining** We first pretrain $f$ with cross-entropy loss, only using labeled data from $\mathcal{D}^l = \mathcal{D}_0$ as an initializer for joint training with the unlabeled domains $\mathcal{D}^u$. To prevent overfitting to $\mathcal{D}^l$, we introduce a noise mixing augmentation technique named NoiseMix. The core idea of this technique is to mix a given sample $\boldsymbol{x}$ with a randomly initialized convolutional layer $\Phi$, such that $\boldsymbol{x} \leftarrow \lambda \Phi(\boldsymbol{x}) + (1 - \lambda)\boldsymbol{x}$, where $\lambda$ is a mixing ratio. The advantage of employing a convolutional layer is that it allows the generation of a noised image without significantly distorting the semantic information of the original image, due to its translational invariance. We apply NoiseMix repeatedly for $P$ steps using a sequence of different random convolutional layers $\Phi_1, \ldots, \Phi_P$. Additional implementation details associated with NoiseMix are provided in Appendix A.

**Prototype-Based Pseudo-Labeling** We present ProtoPL, an extended version of the prototype-based pseudo-labeling method, initially introduced by Liang et al. (2020), to assign pseudo labels to samples from a set of multiple unlabeled domains $\mathcal{D}^u$. In the beginning of every training epoch, we produce a domain-aware class prototype $C_{t,k}$, which indicates the centroid of features for samples in class $k$, for every domain $\mathcal{D}_t$

$$C_{t,k} = \frac{\sum_{\boldsymbol{x}_t \in \mathcal{D}_t} \delta_k(f(\boldsymbol{x}_t))\bar{g}(\boldsymbol{x}_t)}{\sum_{\boldsymbol{x}_t \in \mathcal{D}_t} \delta_k(f(\boldsymbol{x}_t))}, \tag{1}$$

where $\delta_k(\cdot)$ denotes the $k$-th element of a softmax output, and $\bar{g}(\cdot) = g(\cdot)/\|g(\cdot)\|$. Prototypes are generated by calculating a weighted sum of the features as described in Eq. 1, with the weights

representing the probability of belonging to class $k$. Then, each unlabeled sample $\boldsymbol{x}_t$ is pseudo-labeled as

$$\hat{y}(\boldsymbol{x}_t) = \arg\min_k \text{dist}(\bar{g}(\boldsymbol{x}_t), C_{t,k}), \tag{2}$$

where dist is the cosine distance between the feature of sample $\boldsymbol{x}_t$ and the prototypes of its domain $\mathcal{D}_t$. The prototypes are reconstructed based on the new pseudo-labels assigned in Eq. 2 as

$$C_{t,k} = \frac{\sum_{\boldsymbol{x}_d \in \mathcal{D}_t} \mathbb{1}(\hat{y}(\boldsymbol{x}_t) = k)\bar{g}(\boldsymbol{x}_t)}{\sum_{\boldsymbol{x}_t \in \mathcal{D}_t} \mathbb{1}(\hat{y}(\boldsymbol{x}_t) = k)}, \tag{3}$$

Based on the new prototypes, $C_{t,1}, \cdots, C_{t,K}$ for $\mathcal{D}_t$ ($t > 0$), we use Eq. 2 to construct a pseudo-labeled dataset $\widetilde{\mathcal{D}}_t = \{(\boldsymbol{x}_t^{(i)}, \hat{y}_t^{(i)}, \epsilon_t^{(i)})\}_{i=1}^{N_t}$, including the entropy-based uncertainty estimates defined as

$$\epsilon(\boldsymbol{x}_t) = -\sum_k \delta_k(-\text{dist}(\bar{g}(\boldsymbol{x}_t), C_{t,k})/\tau_\epsilon) \log \delta_k(-\text{dist}(\bar{g}(\boldsymbol{x}_t), C_{t,k})/\tau_\epsilon), \tag{4}$$

where $\epsilon(\boldsymbol{x}_t)$ is the uncertainty estimate of sample $\boldsymbol{x}_t$ and $\tau_\epsilon$ is a temperature parameter. The use of this measure will be further explained in a subsequent section, specifically under DomainMix.

In addition to the aforementioned process of ProtoPL, we introduce an ensembling method with random augmentation techniques to reinforce the reliability and robustness of pseudo-labels. Specifically, given a dataset $\mathcal{D}_t$, we generate $R$ randomly augmented datasets $\{A_r\}_{r=1}^R$ and all $N_t \times r$ samples are aggregated to calculate prototypes. Then, pseudo labels are calculated with an ensemble of predictions from different augmentations by

$$\hat{y}(\boldsymbol{x}_t) = \arg\max_k \sum_r \delta_k(-\text{dist}(\bar{g}(A_r(\boldsymbol{x}_t)), C_{t,k})). \tag{5}$$

**Uncertainty-Adaptive Integration of Unlabeled Domains** Since the pretraining process solely depends on samples from a labeled domain, it can be quite challenging to expect the model to generate reliable pseudo-labels for unlabeled domains right from the initial training phase. This necessitates the gradual integration of unlabeled data as the training progresses. To achieve this goal, we introduce a progressive generalization method called DomainMix, where a pseudo-labeled sample is mixed with a labeled sample of the same class based on uncertainty estimates (Eq. 4). By assigning a low mixing ratio for the samples with high uncertainty, we can prevent unreliable samples from being overly involved with training, thereby allowing for a progressive integration of the unlabeled domains. Specifically, given a sample $(\boldsymbol{x}_t, \hat{y}_t, \epsilon_t)$ from pseudo-labeled dataset $\widetilde{\mathcal{D}}_t$, we randomly sample $\boldsymbol{x}_0$ with $y_0 = \hat{y}_t$ from the labeled domain $\mathcal{D}_0$. For a batch of pseudo-labeled samples $\mathcal{B}^u \subset \bigcup_{t=1}^T \mathcal{D}_t$, we can apply the same sampling process for each sample to find a batch of the corresponding labeled samples $\mathcal{B}^l = \text{SampleMatch}(\mathcal{B}^u, \mathcal{D}_0)$. Finally, we mix each pair of the samples $(\boldsymbol{x}^u, \boldsymbol{x}^l)$ from $\mathcal{B}^u$ and $\mathcal{B}^l$ to get a batch of samples $\mathcal{B}^m = \text{DomainMix}(\mathcal{B}^l, \mathcal{B}^u)$ by the following equation:

$$\boldsymbol{x}^m = \lambda\boldsymbol{x}^u + (1 - \lambda)\boldsymbol{x}^l, \tag{6}$$

where $\boldsymbol{x}^m$ is a domain-mixed sample, and $\lambda$ is the mixing ratio. $\lambda$ is primarily determined by the uncertainty estimate $\epsilon$ of the pseudo-labeled sample $\boldsymbol{x}^u$ through $\lambda_\epsilon = \exp(-\epsilon/\tau_\lambda)/(1 + \exp(-\epsilon/\tau_\lambda))$, where $\tau_\lambda$ is a temperature parameter. Note that this equation implies that the uncertainty of labeled samples is assumed to be zero. To address the case where a larger portion of $\boldsymbol{x}^u$ is included in $\boldsymbol{x}^m$ than $\boldsymbol{x}^l$, we assign a random mixing ratio to a sample with $\lambda_\epsilon$ above threshold $\lambda^*$, formulated as

$$\lambda = \begin{cases} \lambda_\text{u} \sim U(0, 1), & \text{if } \lambda_\epsilon > \lambda^*, \\ \lambda_\epsilon, & \text{otherwise}, \end{cases} \tag{7}$$

where $\lambda_\text{u}$ represents a random value sampled from $U(0, 1)$, a uniform random distribution between 0 and 1.

**Loss Function** We first use the conventional cross entropy loss for domain-mixed samples, $\mathcal{L}_\text{CE}(h \circ g; \mathcal{B}^m)$, where $\mathcal{B}^m$ represents a batch of domain-mixed samples. $\mathcal{L}_\text{CE}$ is coupled with

MixUp in the feature level to enhance sample diversity for classification (Zhang et al., 2017). Furthermore, we introduce a contrastive loss for learning domain-invariant representations, called Prototype Merging Loss (PML), defined as

$$\mathcal{L}_{\text{PML}}(g; \mathcal{B}^l \cup \mathcal{B}^u) = - \sum_{(\boldsymbol{x}, y) \in \mathcal{B}^l \cup \mathcal{B}^u} \log \frac{\exp\left(-\operatorname{dist}(\bar{g}(\boldsymbol{x}), \overline{C}_y)\right)}{\sum_k \exp\left(-\operatorname{dist}(\bar{g}(\boldsymbol{x}), \overline{C}_k)\right)}, \tag{8}$$

where $\mathcal{B}^l$ and $\mathcal{B}^u$ represent a batch of labeled samples and a batch of unlabeled samples, respectively, and $\overline{C}_k = \langle C_{t,k} \rangle_t$. The notation $\langle \cdot \rangle_t$ denotes an average over all values of $t$. $\overline{C}_k$ plays a pivotal role as an anchor point, attracting the features of all samples that belong to class $k$, regardless of the domains they originate from. Consequently, PML facilitates the merging of prototypes from different domains into a single point, and the feature extractor $g$ is guided to learn domain-invariant representations, which enhances effective generalization over various domains, encompassing those that are unlabeled.

## 4 EXPERIMENTS

We evaluate the effectiveness of our proposed method, ProUD, by comparing against strong baselines on three datasets: PACS (Li et al., 2017), Digits-DG (LeCun et al., 1998; Ganin et al., 2016; Netzer et al., 2011; Roy et al., 2018), and Office-Home (Venkateswara et al., 2017). Following the experimental setup from Lin et al. (2023), we design our experiments to address SSDG, a representative case of *data inequality problem across domains in which only a sinlge domain is labeled while the rest are unlabeled*.

**Datasets** All three datasets consist of four distinct domains, simulating different sources from which samples are collected. PACS includes Photo (P), Art Painting (A), Cartoon (C), and Sketch (S) with 7 object categories. Digits-DG contains images from MNIST (Mn), MNIST-m (Mm), SVHN (Sv), and SYN-D (Sy) with 10 digits from 0 to 9. Office-Home consists of 65 object categories from Art (Ar), Clipart (Cl), Product (Pr), and Real-World (Rw). We split each source domain dataset ($\mathcal{D}^{\text{train}}$) into training and validation sets with ratio 9:1 for PACS, 8:2 for Digits-DG, and approximately 9:1 for Office-Home.

**Experimental Settings** For each dataset, we conduct experiments for all 12 possible domain combinations across four distinct domains, simulating data inequality problem across domains by including one labeled and two unlabeled source domains ($\mathcal{D}^{\text{train}}$), as well as one unseen test domain ($\mathcal{D}^{\text{test}}$). All experiments are repeated three times with different random seeds (2022/2023/2024), and we report average accuracy obtained from these repetitions. In each experiment, we calculate the average accuracy over the last five epochs. We train the model using the SGD optimizer with mini-batches of 128 samples for all datasets. Further implementation details are provided in Appendix B.

### 4.1 EFFECTIVENESS OF PROUD

**Comparison Baselines** To evaluate the effectiveness of our proposed ProUD method, we benchmark it against a comprehensive set of baselines, using the experiment results reported in Lin et al. (2023). The baselines encompass various methods from the literature on Domain Generalization (DG), Semi-Supervised Learning (SSL), and the integration of Domain Adaptation and Domain Generalization (DA+DG) that are relevant to our SSDG framework. Notably, the state-of-the-art SSDG method, EID, presented by Lin et al. (2023) also serves as one of our baselines. The implementations of these baselines are detailed as follows:

- **Domain Generalization (DG)**: The model employs single-source domain generalization methods such as RSC (Huang et al., 2020), L2D (Wang et al., 2021), and DGvGS (Mansilla et al., 2021), and is trained soley on the labeled source dataset.

- **Semi-Superivsed Learning (SSL)**: The model is trained on both labeled and unlabeled data, disregarding domain distinctions. It employs semi-supervised learning methods, including Mean-Teacher (Tarvainen & Valpola, 2017), MixMatch (Berthelot et al., 2019), and FixMatch (Sohn et al., 2020).

- **Domain Adaptation and Domain Generalization (DA+DG)**: This method offers a direct approach to SSDG and consists of two training phases: pseudo-labeling and domain generalization. Initially, a pseudo-labeling model is trained individually for each unlabeled source domain utilizing unsupervised domain adaptation methods like MCD (Saito et al., 2018) and CDAN+E (Long et al., 2018). Subsequently, the generalization model is trained both on the pseudo-labeled source domains and the true-labeled source domain, leveraging multi-source domain generalization methods such as DDAIG (Zhou et al., 2020a), Mixstyle (Zhou et al., 2021a), and DAELDG (Zhou et al., 2021b).

Table 2: Comparison of the performance between ours and various state-of-the-art methods on the PACS dataset. The bullet • and asterisk * indicate labeled source domain and unseen test domain, respectively. All other domains are treated as the unlabeled source domains. Avg. and Std. respectively stand for average and standard deviation across 12 domain combinations. The best results are highlighted in bold. This representation format is maintained in subsequent tables.

| Type | Method | P• | | | A• | | | C• | | | S• | | | Avg. | Std. |
|---|---|---|---|---|---|---|---|---|---|---|---|---|---|---|---|
| | | A* | C* | S* | P* | C* | S* | P* | A* | S* | P* | A* | C* | | |
| DG | RSC | 66.6 | 27.6 | 38.6 | 93.7 | 68.0 | 65.7 | 83.5 | 69.2 | 76.6 | 47.5 | 43.0 | 65.2 | 62.1 | 18.5 |
| | L2D | 65.2 | 30.7 | 35.4 | 96.1 | 65.7 | 58.0 | 87.3 | 73.5 | 67.9 | 48.2 | 45.9 | 61.8 | 61.3 | 18.6 |
| | DGvGS | 54.2 | 16.6 | 28.5 | 93.8 | 54.7 | 39.7 | 80.3 | 59.5 | 56.7 | 14.3 | 16.2 | 17.2 | 44.3 | 25.5 |
| SSL | MeanTeacher | 54.7 | 36.3 | 33.1 | 91.9 | 65.6 | 38.1 | 80.3 | 60.6 | 58.6 | 38.1 | 33.4 | 54.7 | 53.8 | 18.3 |
| | MixMatch | 35.7 | 16.2 | 24.5 | 87.3 | 62.7 | 47.6 | 43.1 | 47.9 | 50.7 | 26.1 | 46.9 | 52.2 | 45.1 | 18.0 |
| | FixMatch | 66.8 | 34.9 | 25.9 | **96.6** | 72.9 | 67.1 | **91.7** | 76.5 | 69.5 | 36.3 | 35.2 | 56.0 | 60.8 | 22.3 |
| DA | MCD+DDAIG | 71.0 | 53.0 | 55.0 | 93.6 | 64.7 | 65.2 | 90.3 | 74.5 | 69.7 | 47.0 | 45.9 | 45.2 | 64.6 | 15.6 |
| + | MCD+Mixstyle | **75.8** | 61.2 | 50.8 | 95.1 | 67.9 | 64.0 | 89.8 | 78.4 | 68.3 | 48.4 | 44.6 | 53.2 | 66.5 | 15.4 |
| DG | "CDAN+E"+DAELDG | 62.3 | 62.3 | 31.9 | 95.6 | 63.5 | 46.6 | 86.0 | 73.6 | 70.0 | 39.0 | 38.2 | 55.1 | 60.3 | 18.6 |
| SSDG | EID | 75.5 | **71.0** | 64.0 | 94.9 | 71.8 | 67.2 | 84.6 | 77.4 | 72.2 | **67.2** | 66.9 | **72.8** | 73.8 | 8.3 |
| | **ProUD (Ours)** | 73.8 | 63.6 | **74.1** | 91.1 | **75.4** | **76.6** | 86.9 | **78.9** | **78.1** | 63.0 | **68.9** | 70.4 | **75.1** | **8.0** |

Table 3: Comparison of the performance between ours and various state-of-the-art methods on the Digits-DG dataset.

| Type | Method | Mn• | | | Mm• | | | Sv• | | | Sy• | | | Avg. | Std. |
|---|---|---|---|---|---|---|---|---|---|---|---|---|---|---|---|
| | | Mm* | Sv* | Sy* | Mn* | Sv* | Sy* | Mn* | Mm* | Sy* | Mn* | Mm* | Sv* | | |
| DG | RSC | 42.8 | 19.3 | 45.0 | 93.6 | 11.7 | 12.0 | 70.5 | 46.1 | **95.5** | 81.4 | 42.4 | 78.8 | 53.3 | 29.0 |
| | L2D | 57.2 | 28.2 | 53.1 | 97.1 | 12.5 | 25.1 | 72.4 | 52.8 | 94.2 | 80.2 | 45.7 | **80.0** | 58.2 | 26.2 |
| | DGvGS | 13.3 | 12.2 | 19.5 | 88.9 | 11.2 | 16.5 | 57.7 | 24.2 | 88.5 | 68.4 | 25.9 | 67.1 | 41.1 | 29.3 |
| SSL | MeanTeacher | 23.9 | 13.8 | 26.4 | 82.1 | 19.3 | 32.4 | 43.6 | 17.5 | 59.2 | 56.3 | 22.2 | 38.4 | 36.3 | 19.8 |
| | MixMatch | 33.0 | 18.1 | 30.3 | 93.6 | 26.7 | 45.4 | 59.3 | 27.9 | 76.7 | 67.7 | 36.9 | 51.5 | 47.3 | 22.1 |
| | FixMatch | 29.9 | 10.6 | 23.9 | 90.8 | 32.5 | 48.2 | 57.5 | 40.0 | 70.9 | 74.0 | 51.9 | 61.3 | 49.3 | 22.2 |
| DA | MCD+DDAIG | 34.2 | 16.8 | 33.8 | 95.1 | 29.9 | 53.6 | 64.5 | 39.7 | 83.5 | 73.4 | 46.1 | 59.1 | 52.5 | 22.5 |
| + | MCD+Mixstyle | 45.4 | 24.7 | 48.3 | 96.9 | 36.7 | 56.6 | 66.5 | 42.9 | 84.1 | 75.0 | 49.3 | 67.3 | 57.8 | 19.9 |
| DG | "CDAN+E"+DAELDG | 41.2 | 17.2 | 40.6 | 93.8 | 34.6 | 52.4 | 52.7 | 42.3 | 84.2 | 74.6 | 47.9 | 46.7 | 52.4 | 20.8 |
| SSDG | EID | 51.6 | 37.3 | 53.3 | 97.1 | 58.6 | 69.1 | **87.7** | 60.9 | 87.5 | **92.4** | 64.2 | 70.9 | 69.2 | 17.8 |
| | **ProUD (Ours)** | **70.3** | **57.8** | **65.8** | **97.6** | **63.4** | **70.5** | 80.7 | **61.5** | 83.4 | 92.2 | **65.5** | 74.4 | **73.6** | **12.0** |

Table 4: Comparison of the performance between ours and various state-of-the-art methods on the Office-Home dataset.

| Type | Method | Ar• | | | Cl• | | | Pr• | | | Rw• | | | Avg. | Std. |
|---|---|---|---|---|---|---|---|---|---|---|---|---|---|---|---|
| | | Cl* | Pr* | Rw* | Ar* | Pr* | Rw* | Ar* | Cl* | Rw* | Ar* | Cl* | Pr* | | |
| DG | RSC | 39.1 | 49.8 | 61.1 | 36.9 | 53.0 | 53.7 | 35.9 | 38.8 | 61.2 | 53.3 | 45.6 | 72.2 | 50.1 | 10.8 |
| | L2D | 39.6 | 44.8 | 57.5 | 42.2 | 52.6 | 55.7 | 38.5 | 43.0 | 62.3 | 55.0 | 48.3 | 69.3 | 50.7 | 9.2 |
| | DGvGS | 33.4 | 42.9 | 55.4 | 32.6 | 45.0 | 47.0 | 29.8 | 33.2 | 55.0 | 50.8 | 37.9 | 68.0 | 44.3 | 11.1 |
| SSL | MeanTeacher | 35.1 | 50.5 | 60.8 | 39.1 | 51.4 | 54.0 | 35.8 | 34.5 | 62.0 | 54.4 | 43.4 | 72.2 | 49.4 | 11.6 |
| | MixMatch | 40.0 | 51.8 | 62.4 | 43.2 | 57.6 | 58.9 | 42.0 | 38.5 | 63.6 | 55.5 | 43.7 | 72.4 | 52.5 | 10.5 |
| | FixMatch | 41.4 | 55.3 | 64.4 | 44.4 | 57.8 | 57.5 | 44.0 | 42.2 | 65.8 | **57.2** | 45.0 | **73.7** | 54.1 | 10.2 |
| DA | MCD+DDAIG | 42.5 | 54.3 | 63.6 | 42.4 | 53.8 | 56.0 | 40.1 | 37.6 | 59.7 | 48.3 | 43.2 | 69.2 | 50.9 | 9.7 |
| + | MCD+Mixstyle | 44.9 | 55.2 | 65.3 | 45.8 | 56.8 | 59.0 | 42.9 | 42.0 | 63.3 | 52.2 | 46.0 | 69.8 | 53.6 | 9.0 |
| DG | "CDAN+E"+DAELDG | 40.6 | 52.0 | 58.9 | 46.3 | 55.3 | 56.1 | 46.4 | 39.8 | 61.4 | 54.9 | 47.6 | 66.7 | 52.2 | 7.9 |
| SSDG | EID | 48.3 | **59.1** | **66.5** | 47.5 | **60.4** | **61.3** | 46.1 | 47.3 | **66.0** | 53.3 | 48.8 | 69.0 | 56.1 | 8.2 |
| | **ProUD (Ours)** | **48.5** | 57.4 | 65.0 | 46.0 | 57.7 | 59.9 | **47.0** | **49.4** | 65.3 | 55.1 | **52.5** | 71.9 | **56.3** | **7.8** |

**Comparing with State-of-the-Art** Our proposed ProUD method is benchmarked against the baseline methods using the PACS, Digits-DG, and Office-Home datasets. The results for each dataset are detailed in Table 2, 3, and 4, respectively. By averaging performance metrics across 12 distinct domain combinations, as illustrated in these tables, it is evident that ProUD consistently outperforms all baseline models across the datasets. Not only does our approach surpasses the prevailing DG and SSL strategies, but it also exceeds the performance of both DA+DG and EID, which are specifically crafted for the SSDG framework. Importantly, ProUD achieves this improved performance with a single trained model, in contrast to both DA+DG and EID, which necessitate training distinct models for each unlabeled domain to produce pseudo-labels.

Furthermore, it is also crucial to maintain robustness across domain combinations, which represents different scenarios under data inequality problem. This becomes particularly critical when discrepancies between domains impact performance. For instance, as shown in Table 3, RSC exhibits the poorest performance among all baselines for the Mm•/Sy* combination, while it shows the best performance for the Sv•/Sy* combination. The robustness of ProUD is further underscored across Table 2, 3, and 4, where it consistently exhibits the lowest standard deviation over 12 distinct domain combinations across all datasets.

**Training Curve** In Fig. 1, we display the training curve of our model when trained on the PACS dataset, using the labeled source domain as C and the unseen test domain as S. This curve includes the model's accuracy on the unseen domain S, as well as the pseudo-label accuracy and the mixing coefficient $\lambda$ for the unlabeled source domains A and P. Here, $\lambda$ represents the averaged value across all samples within each unlabeled source domain. For the unlabeled source domain A, $\lambda$ begins at a lower value, indicating the high uncertainties inherent in pseudo-labels at the beginning of the training. By assigning a low mixing ratio to samples with high uncertainty, we can limit their influence on training, preventing the potential performance degradation caused by the noisy labels. As training progresses, the model's

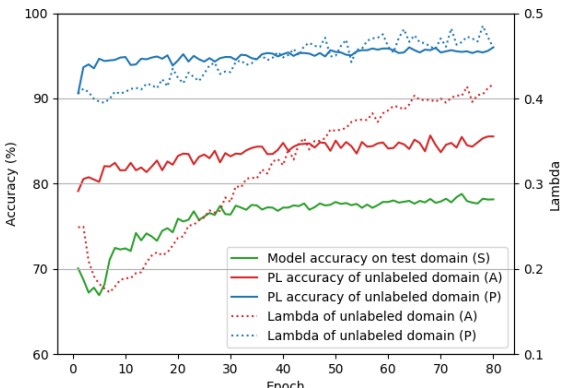

Figure 1: Training curve of ProUD on the PACS dataset, with labeled source domain C, unlabeled source domain A, P, and test domain S. Solid lines and dotted lines represent the accuracy and the value of $\lambda$, respectively.

generalization capabilities strengthen, yielding increasingly precise pseudo-labels with diminishing uncertainties, thereby progressively increasing $\lambda$. In contrast to domain A, unlabeled source domain P starts with a notably higher $\lambda$ value (above 0.4) reflecting more accurate initial pseudo-labels. This enables effective generalization in DomainMix by actively mixing unlabeled data with reliable pseudo-labels at a high ratio.

## 4.2 FURTHER ANALYSIS

**Ablation Study of Uncertainty-Adaptive DomainMix** To validate the effectiveness of Uncertainty-Adaptive DomainMix, we perform an ablation study using the PACS dataset, removing the strategy that determines the mixing coefficient $\lambda$ based on the uncertainty of pseudo-labels. Instead, $\lambda$ is sampled from a uniform distribution ranging from 0 to 1 throughout the training. As shown in Table 5, excluding Uncertainty-Adaptive DomainMix results in decreased performance in the majority of scenarios and amplifies the standard deviation. This underscores its crucial contribution to the model's performance and robustness in ProUD.

**Ablation Study of Prototype Merging Loss** We conduct an ablation study to assess the impact of the Prototype Merging Loss within ProUD, by comparing the performance of models trained both with and without this loss term. The results are shown in Table 5. The exclusion of the loss term results in a decreased average accuracy and an increased standard deviation, underscoring the significance of the Prototype Merging Loss.

Table 5: Ablation study of Uncertainty-Adaptive DomainMix (UAD) and Prototype Merging Loss (PML) on the PACS dataset. Diff. denotes the difference in comparison to our model, ProUD, incorporating both UAD and PML.

| Method | P• | | | A• | | | C• | | | S• | | | Avg. | Std. |
| --- | --- | --- | --- | --- | --- | --- | --- | --- | --- | --- | --- | --- | --- | --- |
| | A* | C* | S* | P* | C* | S* | P* | A* | S* | P* | A* | C* | | |
| ProUD (Ours) | 73.8 | 63.6 | 74.1 | 91.1 | 75.4 | 76.6 | 86.9 | 78.9 | 78.1 | 63.0 | 68.9 | 70.4 | 75.1 | 8.0 |
| w/o UAD | 68.9 | 53.3 | 63.3 | 90.9 | 72.3 | 72.8 | 82.0 | 79.1 | 76.7 | 51.4 | 60.2 | 70.0 | 70.1 | 11.1 |
| Diff. | -4.9 | -10.2 | -11.4 | -0.4 | -3.2 | -3.3 | -5.0 | 0.0 | -1.1 | -11.3 | -9.3 | +0.1 | -5.0 | +3.1 |
| w/o PML | 70.5 | 59.6 | 62.5 | 91.5 | 73.3 | 74.0 | 87.1 | 78.9 | 79.0 | 62.4 | 68.6 | 70.2 | 73.1 | 9.3 |
| Diff. | -3.3 | -3.9 | -12.2 | +0.2 | -2.2 | -2.1 | +0.1 | -0.2 | +1.2 | -0.3 | -0.9 | +0.3 | -2.0 | +1.3 |

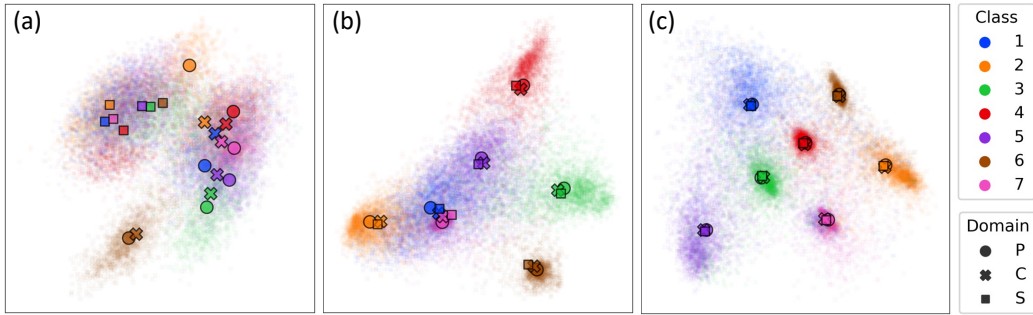

Figure 2: t-SNE visualizations of the learned representations of both samples and domain-aware prototypes using the PACS dataset, in the case where P is the labeled source domain, and C and S are the unlabeled source domains. Different colors and shapes represent distinct classes and domains, respectively. (a), (b), and (c) are produced right after the ProtoPL at epochs 1, 40, and 80, respectively.

**Visualization of Learned Representations**   In Fig. 2, we use t-SNE (van der Maaten & Hinton, 2008) to visualize the learned representations of both samples and domain-aware prototypes, further elucidating the effectiveness of our method. As depicted in Fig. 2 (a), at the beginning of training, class prototypes of the unlabeled source domains (C and S) are closely clustered, while the class prototypes of labeled source domain (P) are relatively well-separated, as a consequence of the model's pretraining on this domain. This behavior points to a fundamental challenge: the model struggles to disentangle domain from class information, resulting into a feature distribution oriented more towards domains than classes. However, by employing the Prototype Merging Loss to merge domain-aware prototypes of the same class, we observe that, as training continues, prototypes of the same class cluster together while distancing from others, illustrated in Fig. 2 (b) and (c). Such observations underscore the effectiveness of the Prototype Merging Loss, affirming its role in inducing domain-invariant features and enhancing the model's classification ability.

## 5   CONCLUSION

In this paper, we address a representative case of the data inequality problem across domains, termed Semi-Supervised Domain Generalization (SSDG), where only one domain is labeled, leaving the others unlabeled. Such a setting mirrors real-world situations, especially when obtaining labeled data from certain domains is considerably more difficult than from others. Moreover, such data inequality not only presents practical challenges but also raises ethical concerns in the design and deployment of machine learning models. To overcome this issue of data inequality, we propose a novel algorithm ProUD to efficiently leverage the potential of samples from unlabeled source domains. The method incorporates prototype-based pseudo labeling (ProtoPL), uncertainty-adaptive integration of unlabeled domains (DomainMix), and contrastive learning for domain-invariant representations (Prototype Merging Loss). Extensive experiments demonstrate that ProUD excels in terms of model performance and robustness across various domain combinations, outperforming state-of-the-art methods in DG, SSL, the integration of DA and DG, and SSDG.

## REPRODUCIBILITY STATEMENT

To ensure the reproducibility of our work, we provide experimental settings in the Experiments section and include additional implementation details in Appendix B. Our source code is available as supplementary materials.

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

APPENDIX

## A    CONVOLUTION-BASED NOISE MIXING AUGMENTATION

In this section, we explain the details of our convolution-based noise mixing augmentation named NoiseMix. We first generate a sequence of convolution layers $(\Phi_p)_{p=1}^P$, where $\Phi_p$ consists of a randomly initialized convolutional layer and an activation function. The random initialization includes Kaiming Uniform parameter initialization He et al. (2015), as well as the random choice of kernel size $k$ and dilation $d$ from uniform distributions. The zero-padding method is used to preserve the size of the image. The hyperbolic tangent function is used as an activation function, followed by normalization.

In Algorithm 2, the pseudocode of NoiseMix is presented. Given a batch of samples $\mathcal{B}_{\text{org}}$, we obtain a batch of noise augmented samples $\mathcal{B}_{\text{aug}}$. Then, we pretrain the model $f$ by concatenating $\mathcal{B}_{\text{org}}$ and $\mathcal{B}_{\text{aug}}$. Besides, we employ NoiseMix in the process of DomainMix. For given $\mathcal{B}_{\text{org}}^l$ and $\mathcal{B}_{\text{org}}^u$, we first generate batches of noise-mixed samples $\mathcal{B}_{\text{aug}}^l$ and $\mathcal{B}_{\text{aug}}^u$. We then apply DomainMix to either $\{(\mathcal{B}_{\text{org}}^l, \mathcal{B}_{\text{org}}^u), (\mathcal{B}_{\text{aug}}^l, \mathcal{B}_{\text{aug}}^u)\}$ or $\{(\mathcal{B}_{\text{org}}^l, \mathcal{B}_{\text{aug}}^u), (\mathcal{B}_{\text{aug}}^l, \mathcal{B}_{\text{org}}^u)\}$ with probability of 0.5.

Throughout this work, we used the same setting of NoiseMix for all the experiments on the three datasets: sequence length $P = 3$, min kernel size $k_{\min} = 1$, max kernel size $k_{\max} = 15$, min dilation $d_{\min} = 1$, max dilation $d_{\max} = 2$, min mixing ratio $\lambda_{\min} = 0.1$, max mixing ratio $\lambda_{\max} = 0.3$, inversion probability $p = 0.2$, and noise strength $\sigma = 0.01$.

---

**Algorithm 2** NoiseMix

---

**Input:** Sample $\boldsymbol{x}_{\text{org}}$, a sequence of randomly initialized convolution layers $\Phi_1, \dots, \Phi_P$, and a sequence of randomly sampled mixing ratio $\lambda_1, \dots, \lambda_P$.
$\boldsymbol{x} = \text{RandomInvert}(\boldsymbol{x}_{\text{org}})$
**for** $p = 1$ to $P$ **do**
    Sample a Gaussian noise $\mathbf{n} \sim \mathcal{N}(\mathbf{0}, \sigma \boldsymbol{I})$
    $\tilde{\boldsymbol{x}} = \boldsymbol{x} + \mathbf{n}$
    $\boldsymbol{x} = \lambda_p \Phi_p(\tilde{\boldsymbol{x}}) + (1 - \lambda_p)\tilde{\boldsymbol{x}}$
**end for**
$\boldsymbol{x}_{\text{aug}} = \text{RandomInvert}(\text{Normalize}(\text{Sigmoid}(\boldsymbol{x})))$
**return** $\boldsymbol{x}_{\text{aug}}$

---

## B    IMPLEMENTATION DETAILS

**Experimental Settings**    We conduct experiments on three datasets: PACS, Digits-DG, and Office-Home, training the model for 80, 800, and 60 epochs each. For optimization, we use the SGD optimizer, with a momentum of 0.9 and weight decays of 0.01, 0.005, and 0.0001 corresponding to each dataset. The learning rate is set to 0.001 for PACS and Office-Home, and 0.01 for Digits-DG.

For PACS and Office-Home, we use simple augmentation techniques including random translation, horizontal flipping, color jittering, and grayscaling. The same augmentations are applied to the Digits-DG dataset, except for horizontal flipping. In ProtoPL, each dataset utilizes three different augmentations for ensemble learning ($r$), while the temperature parameter ($\tau_\epsilon$) is 0.03, 0.1, and 0.02 for PACS, Digits-DG, and Office-Home, respectively. For DomainMix, the threshold of $\lambda$ ($\lambda^*$) is set at 0.35, and the temperature parameter ($\tau_\lambda$) is 0.5, consistent across all datasets. The balancing parameter in the loss function ($\alpha$) is set to 0.5. We set the Mixup hyperparameter to 2.0 for PACS and Office-Home, and to 0.4 for Digits-DG.

**Network Architectures**    The network of our model is composed of a feature extractor and a classifier. We utilize ResNet-18 (He et al., 2016) as the feature extractor for both PACS and Office-Home. For Digits-DG, we use a different backbone consisting of four convolutional layers. ResNet-18 is initialized with the weights pretrained on ImageNet (Deng et al., 2009). The classifier consists of a single fully connected layer with weight normalization for all three datasets.

# C  FURTHER ANALYSIS ON THE IMPACT OF PROTOTYPE MERGING LOSS

In this section, we assess the impact of Prototype Merging Loss (PML) through t-SNE visualization. In Fig. 3, we observe that the performance of our algorithm without the implementation of PML. In contrast to the distinct clustering observed in Fig. 2, the samples are only loosely clustered around their respective prototypes here, resulting in more ambiguous cluster boundaries than those seen with PML. Even after 80 epochs, several prototypes still remain intermingled, demonstrating a lack of distinct discrimination. These observations underscore the vital role of PML, as formulated in Eq. 8. PML enhances the algorithm's ability to learn domain-invariant representations by effectively merging prototypes of the same class from diverse domains, while simultaneously ensuring their separation from prototypes of different classes.

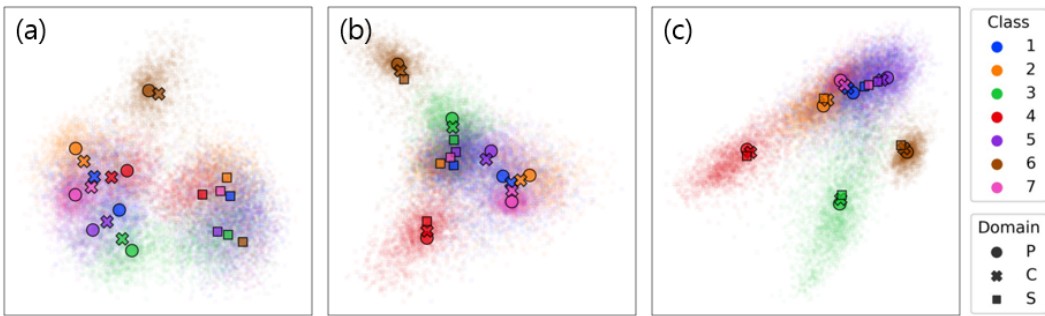

Figure 3: t-SNE visualizations of the learned representations of both samples and domain-aware prototypes when the Prototype Merge Loss is removed. P is the labeled source domain, and C and S are the unlabeled source domains from the PACS dataset. Different colors and shapes represent distinct classes and domains, respectively. (a), (b), and (c) are produced right after the ProtoPL at epochs 1, 40, and 80, respectively.

