# OpenReview forum: "Overcoming Data Inequality across Domains with Semi-Supervised Domain Generalization"
_ICLR.cc/2024/Conference — ICLR 2024 Conference Withdrawn Submission_

### Official Review · Reviewer_oRJe · 2023-10-17

**Soundness:** 2 fair
**Presentation:** 3 good
**Contribution:** 3 good
**Rating:** 6
**Confidence:** 4

**Summary:**

This paper addresses a representative case of data inequality problem across domains termed Semi-Supervised Domain Generalization (SSDG), in which only one domain is labeled while the rest are unlabeled. It proposes a novel algorithm, ProUD, designed for progressive generalization across domains by leveraging domain-aware prototypes and uncertaintyadaptive mixing strategies.

**Strengths:**

1. The direction of the research in the paper is meaningful and has the potential to generate positive impact.

2. The description of the algorithm in the paper is clear, making it easy to read and reproduce.

3. The description of experimental settings in the appendix is detailed.

**Weaknesses:**

1. The paper is not well-written, as the algorithm lacks both theoretical support and adequate explanation, making it difficult to understand the authors' rationale behind the algorithm design.
2. The experiment, as it stands currently, is not sufficiently refined for the following reasons: 1). The datasets used in the experiment are all simple and small-scale; 2). On some datasets, the performance improvement compared to EID is relatively small, and the choice of random seed is not general enough; 3). There is only one partition for each dataset, and the proportions are not uniform (9:1 and 8:2). The author did not provide an explanation for the choice of different partitions.

**Questions:**

1. The algorithm uses prototypes obtained from soft labels when acquiring pseudo-labels (Equation 1), and prototypes obtained from hard labels are used in subsequent calculations of uncertainty and loss functions (Equation 3). What is the reason for these choices, and what is the difference in effectiveness between the two types of prototypes?

2. How was Equation 7 derived, and what is the rationale behind choosing it?

3. What is the improvement brought about by data augmentation, and how would the results compare if all algorithms used the same number of data augmentations in the comparison?

---

> ### Author Response · Authors · 2023-11-20
> **A Response to Questions**
>
> We appreciate the reviewer's recognition of the meaningful direction and potential impact of our research, as well as the clarity of the algorithm's description. We are eager to clarify the points raised in your questions and concerns regarding the experimental design and rationale behind our algorithm.
>
> >The algorithm uses prototypes obtained from soft labels when acquiring pseudo-labels (Equation 1), and prototypes obtained from hard labels are used in subsequent calculations of uncertainty and loss functions (Equation 3). What is the reason for these choices, and what is the difference in effectiveness between the two types of prototypes?
>
> The alternating use of soft and hard labels is followed by the previous work [1]. Our contribution involves extending the pseudo-labeling method of Liang et al. to be effective in Semi-Supervised Domain Generalization (SSDG) by leveraging domain-aware class prototypes.
>
> >How was Equation 7 derived, and what is the rationale behind choosing it?
>
> The definition of  $\lambda_\epsilon = \exp (-\epsilon/\tau_{\lambda})/(1+\exp(-\epsilon/\tau_{\lambda}))$ assumes that the uncertainty of the “labeled” samples to be zero, which is the minimum value defined by Eq. 4. However, due to the inherent noisiness of the real-world data, it is not feasible for the uncertainty of unlabeled domains to be exactly zero. To promote more diverse augmentation, we introduce the threshold $\lambda^*$ in Eq. 7, and in case $\lambda > \lambda^*$, we randomly choose the mixing ratio from a uniform distribution between 0 and 1, so that it is possible to mix with higher portion of unlabeled samples when its uncertainty is below a certain level of threshold. We found that the value could be fixed to 0.35, as the same value consistently yielded good results across all datasets and experiments.
>
> >What is the improvement brought about by data augmentation, and how would the results compare if all algorithms used the same number of data augmentations in the comparison?
>
> We conducted an additional experiment on the Digits-DG dataset, reducing the number of data augmentations from 3 to 1. The average accuracy decreased by 0.24, yet this performance still outperforms all the baselines.
>
> [1] Jian Liang, Dapeng Hu, and Jiashi Feng. Do we really need to access the source data? source hypothesis transfer for unsupervised domain adaptation. In International conference on machine learning, pp. 6028–6039. PMLR, 2020.

---

> ### Author Response · Authors · 2023-11-20
> **A Response to Weakness 1**
>
> >The paper is not well-written, as the algorithm lacks both theoretical support and adequate explanation, making it difficult to understand the authors' rationale behind the algorithm design.
>
> In order to clarify the rationale behind the algorithm design, we would like to provide a theoretical background behind our work. Semi-supervised domain generalization, a specialized form of semi-supervised learning, is particularly relevant in scenarios where labeled and unlabeled data exhibit differing distributions or domains. To address this issue, we utilize the measure of uncertainty inherent in each pseudo-label to modulate the involvement of each sample in the training process.
>
> Specifically, we employ MixUp between labeled and unlabeled data, applying a reduced mixing ratio in domains with higher uncertainty. By assigning a low mixing ratio to samples with high uncertainty, we prevent these unreliable samples from overly influencing the training process, thereby allowing for a more progressive integration of unlabeled domains. This strategy successfully addresses the scalability issues found in the previous state-of-the-art method [1], as we do not require a distinct DA model for each domain.
>
> However, directly applying MixUp to domain generalization can be problematic, as it does not distinguish between domain and class information. This entanglement of domain and class knowledge can result in performance degradation, as identified in [2]. To circumvent this issue, we first limit the application of MixUp to images from identical classes (DomainMix) and implement the Prototype Merging Loss, which reduces intra-class distance across varying domains to enhance generalization capabilities, while simultaneously, maintaining a clear separation between data points of different classes for effective discrimination. This approach prevents the entanglement of domain and class knowledge, enabling the effective learning of domain-invariant features that perform class discrimination across datasets from different domains.
>
> [1] Luojun Lin, Han Xie, Zhishu Sun, Weijie Chen, Wenxi Liu, Yuanlong Yu, and Lei Zhang. Semi-supervised domain generalization with evolving intermediate domain. arXiv preprint arXiv:2111.10221, 2023.
>
> [2] Wang Lu, Jindong Wang, Han Yu, Lei Huang, Xiang Zhang, Yiqiang Chen, and Xing Xie. Fixed: Frustratingly easy domain generalization with mixup. arXiv preprint arXiv:2211.05228, 2022.

---

> ### Author Response · Authors · 2023-11-20
> **A Response to Weakness 2**
>
> >The datasets used in the experiment are all simple and small-scale
>
> While they seem relatively simple and small-scale, PACS, Digits-DG, or Office-home have been used as important benchmark datasets in many of the recent papers published in major ML conferences, such as CVPR([1,2,3]), ICCV([4,5,6]), or ICLR([7,8]).
>
> >On some datasets, the performance improvement compared to EID is relatively small, and the choice of random seed is not general enough
>
> Although the margin of performance improvement may seem relatively small, we would like to emphasize that the major advantage of ProUD over EID lies in its scalability. EID requires separate domain adaptation models and their respective training processes for each unlabeled source domains, to address SSDG. This imposes a significant limitation on its practical application to SSDG problems with a large number of source domains. The same challenge applies to attempts to exploit different combinations of SoTA models for DA and DG. Therefore, our single model-based approach offers a significant practical advantage in handling numerous source domains in a scalable manner.
>
> We would also like to clarify that the choice of random seed has been made with no intention of influencing, or any relation to, the experimental results. Please note that the standard deviations of the average accuracy over different seeds are marginal, and the lower bounds of the average accuracy $(\mu- \sigma)$ still outperform all the baselines.
> | Seed |  PACS | Office-Home | Digits-DG |
> |:----:|:-----:|:-----------:|:---------:|
> | 2022 | 75.08 |    56.33    |   74.37   |
> | 2023 | 75.09 |    56.24    |   74.26   |
> | 2024 | 75.02 |    56.28    |   74.10   |
> | Avg. | 75.06 |    56.28    |   74.24   |
> | Std. |  0.03 |     0.04    |    0.56   |
>
> >There is only one partition for each dataset, and the proportions are not uniform (9:1 and 8:2). The author did not provide an explanation for the choice of different partitions.
>
> For the partitioning of each dataset, we followed EID for fair comparison.
>
> [1] Shiqi Lin, Zhizheng Zhang, Zhipeng Huang, Yan Lu, Cuiling Lan, Peng Chu, Quanzeng You, Jiang Wang, Zicheng Liu, Amey Parulkar, Viraj Navkal, and Zhibo Chen. "Deep frequency filtering for domain generalization." Proceedings of the IEEE/CVF Conference on Computer Vision and Pattern Recognition. 2023.
>
> [2] Wenke Huang, Mang Ye, Zekun Shi, He Li, and Bo Du. "Rethinking federated learning with domain shift: A prototype view." Proceedings of the IEEE/CVF Conference on Computer Vision and Pattern Recognition. 2023.
>
> [3] Jin Chen, Zhi Gao, Xinxiao Wu, and Jiebo Luo. "Meta-causal Learning for Single Domain Generalization." Proceedings of the IEEE/CVF Conference on Computer Vision and Pattern Recognition. 2023.
>
> [4] Sheng Cheng, Tejas Gokhale, and Yezhou Yang. "Adversarial Bayesian Augmentation for Single-Source Domain Generalization." Proceedings of the IEEE/CVF International Conference on Computer Vision. 2023.
>
> [5] Xiran Wang, Jian Zhang, and Lei Qi. "Generalizable Decision Boundaries: Dualistic Meta-Learning for Open Set Domain Generalization." Proceedings of the IEEE/CVF International Conference on Computer Vision. 2023.
>
> [6] Chaoqi Chen, Luyao Tang, Leitian Tao, Hong-Yu Zhou, Yue Huang, Xiaoguang Han, and Yizhou Yu. "Activate and Reject: Towards Safe Domain Generalization under Category Shift." Proceedings of the IEEE/CVF International Conference on Computer Vision. 2023.
>
> [7] Ziqiao Wang, and Yongyi Mao. "Information-Theoretic Analysis of Unsupervised Domain Adaptation." The Eleventh International Conference on Learning Representations. 2022.
>
> [8] Yan Yan, and Yuhong Guo. "Partial Label Unsupervised Domain Adaptation with Class-Prototype Alignment." The Eleventh International Conference on Learning Representations. 2022.

---

> > ### Comment · Reviewer_oRJe · 2023-11-22
> >
> > Thank you for your thoughtful response, which addressed some of my concerns. For me, this is a paper teetering on the edge of acceptance, and it's quite nuanced. I'm inclined to give it a score of 5.5. Elevating it to a 6 would be a way to commend the author for the earnest attitude displayed during the rebuttal process. The final decision, however, will depend on a comparison with the quality of other submissions. The paper's strength lies in its meaningful research direction and overall well-written content. Areas for improvement include a lack of clear innovation in the algorithm, rigid details in algorithm design that lack persuasive power, and a relatively modest practical performance improvement. The paper falls short of reaching an outstanding level of appeal.

---

> ### Author Response · Authors · 2023-11-23
>
> We appreciate your decision to raise the score to 6. We are pleased that our response has addressed some of your concerns. Thank you for your thoughtful feedback.

---

### Official Review · Reviewer_Rcny · 2023-10-30

**Soundness:** 1 poor
**Presentation:** 2 fair
**Contribution:** 1 poor
**Rating:** 5
**Confidence:** 3

**Summary:**

The paper proposes a new problem setting across domains termed Semi-Supervised Domain Generalization (SSDG), in which only one domain is labeled while the rest are unlabeled. The paper proposed a semi-supervised learning method called ProUD, by leveraging domain-aware prototypes, uncertainty adaptive mixing strategies, and pseudo labels. The authors conduct experiments on three datasets (PACS, Digits-DG, Office-Home) to demonstrate the effectiveness of ProUD.

**Strengths:**

- The paper proposes an interesting setting Semi-Supervised Domain Generalization (SSDG), in which only one domain is labeled while the rest are unlabeled. In the introduction, the paper lists some data inequality scenarios where SSDG may be used (Table 1).
- It is a good try to introduce semi-supervised learning to the domain generalization community, e.g., how to construct pseudo labels.

**Weaknesses:**

- The major concern, from my perspective, is the experiments section. (1) The paper misses many important benchmark datasets, such as VLCS, TerraInc, and DomainNet. See detail in DomainBed [1]. In particular, DomainBed is an important benchmark. (2) The paper fails to compare lots of state-of-the-art methods. The latest works compared in the paper are published in 2021. There are lots of good works in 2022 and 2023 that should be included, such as [2,3,4] and many more. (3) The results in Table 2, Table 3, and Table 4 cannot even show that ProUD outperforms EID by a large margin. I am not convinced that the proposed method ProUD is a useful method.
- Some minor weaknesses. (1) The proposed methods need domain labels for domain-aware prototypes. This will restrict the methods when applied to the application. It would be good to consider whether there is a way to generalize the method to be domain label-free. (2) The methods are simple and trivial. In my understanding, PML loss (Equation 8) is pretty similar to SupCon [5]. (3) It would be good if the paper could introduce some theoretical analysis to give more insights or intuition about the ProUD.

[1] Gulrajani, Ishaan, and David Lopez-Paz. In search of lost domain generalization. arXiv 2020.

[2] Junbum Cha, Kyungjae Lee, Sungrae Park, and Sanghyuk Chun. Domain generalization by mutual-information regularization with pre-trained models. ECCV 2022.

[3] Junbum Cha, Sanghyuk Chun, Kyungjae Lee, Han-Cheol Cho, Seunghyun Park, Yunsung Lee, and Sungrae Park. Swad: Domain generalization by seeking flat minima. NeurIPS 2021.

[4] Bo Li, Yifei Shen, Jingkang Yang, Yezhen Wang, Jiawei Ren, Tong Che, Jun Zhang, and Ziwei Liu. Sparse mixture-of-experts are domain generalizable learners. ICLR 2023.

[5] Khosla, Prannay, Piotr Teterwak, Chen Wang, Aaron Sarna, Yonglong Tian, Phillip Isola, Aaron Maschinot, Ce Liu, and Dilip Krishnan. Supervised contrastive learning. NeurIPS 2020.

**Questions:**

See the weakness above.

---

> ### Author Response · Authors · 2023-11-20
>
> Firstly, we would like to express our gratitude for your critical reviews and valuable insights aimed at improving our work. We acknowledge the concerns raised regarding certain aspects of our research. In the following comments, we will provide a detailed, point-by-point response, which we hope will ensure a comprehensive understanding of our methods and contributions.
>
> >The paper misses many important benchmark datasets, such as VLCS, TerraInc, and DomainNet. See detail in DomainBed [1]. In particular, DomainBed is an important benchmark.
>
> PACS, Digits-DG, and Office-home have been predominantly used as key benchmarks without including VLCS, TerraInc, or DomainNet, in many recent papers published in major ML conferences,  such as CVPR [6,7,8], ICCV [9,10,11], or ICLR [12,13].
>
> >The paper fails to compare lots of state-of-the-art methods. The latest works compared in the paper are published in 2021. There are lots of good works in 2022 and 2023 that should be included, such as [2,3,4] and many more.
>
> SSDG is a new problem setting that has not been extensively discussed previously.  The works [2,3,4] you mentioned are effective SoTA methods in DG. However, those methods are primarily multi-source domain generalization approaches, which are not directly applicable to SSDG, due to the absence of labels on most source domains. Instead, here we present an additional comparison of our method with the latest SoTA methods on single-source DG on the PACS dataset, including the one published in CVPR 2023 [8], thereby demonstrating the effectiveness of our method:
>
> |    Method    |   P  |   A  |   C  |   S  | Avg. |
> |:------------:|:----:|:----:|:----:|:----:|:----:|
> | RSC+ASR [14] | 54.6 | 76.7 | 79.3 | 61.6 | 68.1 |
> |   MCL [8]   | 59.6 | 77.1 | 80.1 | 62.6 | 69.9 |
> |     Ours     | 70.5 | 81.0 | 81.3 | 67.4 | 75.1 |
>
> >The results in Table 2, Table 3, and Table 4 cannot even show that ProUD outperforms EID by a large margin. I am not convinced that the proposed method ProUD is a useful method.
>
> Beyond its superior performance, it is important to highlight ProUD's major advantage over EID [15] in terms of scalability. EID requires a separate domain adaptation model and training process for each unlabeled source domain, which becomes impractical for SSDG problems involving many source domains. This limitation also applies to attempts at combining different SoTA models for DA and DG. In contrast, ProUD utilizes a single model capable of handling multiple source domains through its progressive generalization algorithm in a scalable manner.

---

> ### Author Response · Authors · 2023-11-20
>
> >The proposed methods need domain labels for domain-aware prototypes. This will restrict the methods when applied to the application. It would be good to consider whether there is a way to generalize the method to be domain label-free.
>
> SSDG is a new problem setting that assumes the presence of implicit domain labels. This setting is realistic in many scenarios as illustrated in Table 1. For instance, in biomedical imaging, lots of hospitals (domains) obtain medical image data but only few of them undergo additional process for labeling the images. Our method aims to present how to effectively utilize data from unlabeled domains along with implicit domain labels in a scalable manner. Existing methods for semi-supervised learning can also be applied for SSDG, but it is difficult for them to be as effective due to their inability to utilize available domain labels, as presented in our experimental results (Table 2,3, and 4).
>
> >The methods are simple and trivial. In my understanding, PML loss (Equation 8) is pretty similar to SupCon [5].
>
> Contrastive learning itself is a conventional approach, and various studies have derived their own methods for applying this framework to their specific problem settings. The Supervised Contrastive Loss in SupCon [5] is designed to attract features of samples within the same class to each other, yet it lacks consideration for different domains, which is crucial in SSDG.
>
> On the other hand, PML aims to attract the features of samples towards their corresponding $\bar{C}_k$, which represents the average of the same class prototypes from all source domains. Merging the class prototypes from different domains not only enhances the model’s generalization ability by making features domain-invariant, as visualized in Fig. 2 (further analysis is added in Appendix C of the revised manuscript), but also offers a significant advantage in representing under-represented domains. This approach prevents representation bias caused by the uneven distribution of samples across domains.
>
> >It would be good if the paper could introduce some theoretical analysis to give more insights or intuition about the ProUD.
>
> In order to clarify the intuition about the ProUD, we would like to provide a theoretical rationale behind our work. Semi-supervised domain generalization, a specialized form of semi-supervised learning, is particularly relevant in scenarios where labeled and unlabeled data exhibit differing distributions or domains. To address this issue, we utilize the measure of uncertainty inherent in each pseudo-label to modulate the involvement of each sample in the training process.
>
> Specifically, we employ MixUp between labeled and unlabeled data, applying a reduced mixing ratio in domains with higher uncertainty. By assigning a low mixing ratio to samples with high uncertainty, we prevent these unreliable samples from overly influencing the training process, thereby allowing for a more progressive integration of unlabeled domains. This strategy successfully addresses the scalability issues found in the previous state-of-the-art method [15], as we do not require a distinct DA model for each domain.
>
> However, directly applying MixUp to domain generalization can be problematic, as it does not distinguish between domain and class information. This entanglement of domain and class knowledge can result in performance degradation, as identified in [16]. To circumvent this issue, we first limit the application of MixUp to images from identical classes (DomainMix) and implement the Prototype Merging Loss, which reduces intra-class distance across varying domains to enhance generalization capabilities, while simultaneously, maintaining a clear separation between data points of different classes for effective discrimination. This approach prevents the entanglement of domain and class knowledge, enabling the effective learning of domain-invariant features that perform class discrimination across datasets from different domains.

---

> ### Author Response · Authors · 2023-11-20
>
> [1] Gulrajani, Ishaan, and David Lopez-Paz. In search of lost domain generalization. arXiv 2020.
>
> [2] Junbum Cha, Kyungjae Lee, Sungrae Park, and Sanghyuk Chun. Domain generalization by mutual-information regularization with pre-trained models. ECCV 2022.
>
> [3] Junbum Cha, Sanghyuk Chun, Kyungjae Lee, Han-Cheol Cho, Seunghyun Park, Yunsung Lee, and Sungrae Park. Swad: Domain generalization by seeking flat minima. NeurIPS 2021.
>
> [4] Bo Li, Yifei Shen, Jingkang Yang, Yezhen Wang, Jiawei Ren, Tong Che, Jun Zhang, and Ziwei Liu. Sparse mixture-of-experts are domain generalizable learners. ICLR 2023.
>
> [5] Khosla, Prannay, Piotr Teterwak, Chen Wang, Aaron Sarna, Yonglong Tian, Phillip Isola, Aaron Maschinot, Ce Liu, and Dilip Krishnan. Supervised contrastive learning. NeurIPS 2020.
>
> [6] Shiqi Lin, Zhizheng Zhang, Zhipeng Huang, Yan Lu, Cuiling Lan, Peng Chu, Quanzeng You, Jiang Wang, Zicheng Liu, Amey Parulkar, Viraj Navkal, and Zhibo Chen. "Deep frequency filtering for domain generalization." *Proceedings of the IEEE/CVF Conference on Computer Vision and Pattern Recognition*. 2023.
>
> [7] Wenke Huang, Mang Ye, Zekun Shi, He Li, and Bo Du. "Rethinking federated learning with domain shift: A prototype view." *Proceedings of the IEEE/CVF Conference on Computer Vision and Pattern Recognition*. 2023.
>
> [8] Jin Chen, Zhi Gao, Xinxiao Wu, and Jiebo Luo. "Meta-causal Learning for Single Domain Generalization." *Proceedings of the IEEE/CVF Conference on Computer Vision and Pattern Recognition*. 2023.
>
> [9] Sheng Cheng, Tejas Gokhale, and Yezhou Yang. "Adversarial Bayesian Augmentation for Single-Source Domain Generalization." *Proceedings of the IEEE/CVF International Conference on Computer Vision*. 2023.
>
> [10] Xiran Wang, Jian Zhang, and Lei Qi. "Generalizable Decision Boundaries: Dualistic Meta-Learning for Open Set Domain Generalization." *Proceedings of the IEEE/CVF International Conference on Computer Vision*. 2023.
>
> [11] Chaoqi Chen, Luyao Tang, Leitian Tao, Hong-Yu Zhou, Yue Huang, Xiaoguang Han, and Yizhou Yu. "Activate and Reject: Towards Safe Domain Generalization under Category Shift." *Proceedings of the IEEE/CVF International Conference on Computer Vision*. 2023.
>
> [12] Ziqiao Wang, and Yongyi Mao. "Information-Theoretic Analysis of Unsupervised Domain Adaptation." *The Eleventh International Conference on Learning Representations*. 2022.
>
> [13] Yan Yan, and Yuhong Guo. "Partial Label Unsupervised Domain Adaptation with Class-Prototype Alignment." *The Eleventh International Conference on Learning Representations*. 2022.
>
> [14] Xinjie Fan, Qifei Wang, Junjie Ke, Feng Yang, Boqing Gong, and Mingyuan Zhou. Adversarially adaptive normalization for single domain generalization. *Proceedings of the IEEE/CVF Conference on Computer Vision and Pattern Recognition*. 2021.
>
> [15] Luojun Lin, Han Xie, Zhishu Sun, Weijie Chen, Wenxi Liu, Yuanlong Yu, and Lei Zhang. Semi-supervised domain generalization with evolving intermediate domain. arXiv preprint arXiv:2111.10221, 2023.
>
> [16] Wang Lu, Jindong Wang, Han Yu, Lei Huang, Xiang Zhang, Yiqiang Chen, and Xing Xie. Fixed: Frustratingly easy domain generalization with mixup. arXiv preprint arXiv:2211.05228, 2022.

---

> ### Author Response · Authors · 2023-11-23
> **With the hope that our response addresses your concerns**
>
> Dear Reviewer Rcny,
>
> As the discussion period draws to a close, we eagerly await your response. We greatly appreciate your valuable contributions to the review and enhancement of our paper.
>
> We have provided detailed responses to each of your concerns. Please review our responses again and kindly let us know whether they adequately address your concerns and if our explanations are heading in the right direction. Any additional feedback would be highly appreciated.
>
> Kind regards,
>
> Authors of Paper1702

---

> > ### Comment · Reviewer_Rcny · 2023-11-23
> >
> > Thanks for the detailed rebuttal. It partially fixed my concern, and I updated my score from 3 to 5. Please include a full comparison with MCL in future revisions.
> >
> > The reason I raised my score is that I accept the SSDG with domain labels as a meaningful problem setting and the proposed method ProUD can beat MCL, although there are only limited baselines in this direction.
> >
> > The reason I still do not tend to accept:
> > - The method lacks novelty as Reviewer sXNN comments.
> > - The paper can be stronger if authors evaluate different baselines and their methods on DomainNet.
> > - The theoretical analysis part is vague. See a related sample paper below [1] for reference.
> >
> > [1] Deng, Yihe, Yu Yang, Baharan Mirzasoleiman, and Quanquan Gu. Robust Learning with Progressive Data Expansion Against Spurious Correlation. NeurIPS 2023.

---

> > > ### Author Response · Authors · 2023-11-23
> > >
> > > We are grateful for your decision to raise the score to 5 and are pleased to know that our response has addressed some of your concerns. In accordance with your suggestions, we will incorporate a full comparison with MCL in the final version of our paper and review the sample paper you kindly provided to enhance our theoretical analysis. Thank you for your thoughtful feedback and suggestions.

---

### Official Review · Reviewer_sXNN · 2023-11-01

**Soundness:** 2 fair
**Presentation:** 2 fair
**Contribution:** 2 fair
**Rating:** 5
**Confidence:** 3

**Summary:**

This paper studies a practically important problem called Semi-Supervised Domain Generalization (SSDG), where the multiple source domains contain both labeled samples and unlabelled samples. Its goal is to generalize the model trained on the source domains to an unseen target domain. To address this issue, the authors propose a novel algorithm called ProUD, which leverages domain-aware prototypes and uncertainty-adaptive mixing strategies.

**Strengths:**

1. **[The problem of this paper is critical in practice.]** In the previous studies under domain generalization, they always assume that labels of multiple source domains are available. However, it is sometimes infeasible to obtain such perfect source domains, which gives rise to the importance of SSDG.
2. **[This paper is well written and easy to follow.]** Background, motivation, details about the proposed method,  and experiments are well introduced.

**Weaknesses:**

1. **[The proposed method is lack of novelty.]** Essentially, the proposed method is still a combination of DA +DG. The step for assigning pseudo labels for unlabelled source domains can be regarded as DA, while the step for learning domain-invariant representations via a contrastive loss can be regarded as DG. Tools used in each step are also widely used in methods for DA and DG basically
2. **[Motivation of this proposed method is unclear.]** It is unclear why the authors propose this method. For example, if there exists a research gap for current studies on SSGD?
3. **[The propsed method is not explored deeply.]** Firstly, this paper does not provide a theoretical analysis to certify the effectiveness of the proposed method. Furthermore, it is unclear how the accuracy of pseudo-labels affects the final performance. In detail, in Equation (6), you mix labeled and unlabeled samples with the same class up, whose performance may heavily rely on the quality of the pseudo labels. Additionally, in the t-SNE Visualization, I want to know if Equation (8) contributes to the representation most.

**Questions:**

1. How do you choose the value of the threshold $\lambda^*$?
2. Can you provide insights to answer the questions in Weakness 2 and Weakness 3?

---

> ### Author Response · Authors · 2023-11-20
> **A Response to Questions**
>
> Thank you for your insightful comments and constructive criticisms regarding our paper. We appreciate your recognition of the practical importance of SSDG and the clarity of presentation in our work. We'd like to address the points you raised in the 'Questions' section and resolve the concerns you brought up under 'Weaknesses.’
>
> >$\textit{How do you choose the value of the threshold $\lambda^∗$?}$
>
> We chose the value of the threshold $\lambda^∗$ from the following set of values {0.3, 0.35, 0.4, 0.45} by evaluating on validation dataset. However, we found that the value $\lambda^∗$ could be fixed to 0.35, as the same value consistently yielded good results across all datasets and experiments.
>
> >$\textit{Can you provide insights to answer the questions in Weakness 2 and Weakness 3?}$
>
> >  $\textbf{A Response to Weakness 2 (along with Weakness 1)}$
>
> >   [The proposed method is lack of novelty.] Essentially, the proposed method is still a combination of DA +DG. The step for assigning pseudo labels for unlabelled source domains can be regarded as DA, while the step for learning domain-invariant representations via a contrastive loss can be regarded as DG. Tools used in each step are also widely used in methods for DA and DG basically
>
> >   [Motivation of this proposed method is unclear.] It is unclear why the authors propose this method. For example, if there exists a research gap for current studies on SSGD?
>
> The current state-of-the-art in SSDG [1], requires as many separate domain adaptation models and training processes as there are unlabeled source domains. This requirement imposes a significant limitation on its practical application to SSDG problems with a large number of source domains. The same obstacle applies to any attempts to exploit different combinations of state-of-the-art models for DA and DG.
>
> However, ProUD’s contribution lies in overcoming the limitations of previous methods that encounter scalability problems. ProUD’s progressive generalization algorithm allows for handling multiple unlabeled source domains simultaneously with a single, scalable model through the effective use of a measure of uncertainty based on domain-aware prototypes.
>
> Furthermore, our DomainMix method not only modulates the involvement of samples from each unlabeled domain based on their uncertainty but also enhances the diversity of samples from under-represented unlabeled domains by augmenting them through an adaptive mix with labeled samples. The Prototype Merging Loss (PML), another key component of ProUD, constructs mean prototypes ($\bar{C}_k$) by assigning equal weights to the same class prototypes from each domain, irrespective of the number of samples in each domain. This approach offers a significant advantage in representing under-represented domains, preventing representation bias caused by uneven sample distribution across domains.
>
> > $\textbf{A Response to Weakness 3}$
>
> >  [The proposed method is not explored deeply.] Firstly, this paper does not provide a theoretical analysis to certify the effectiveness of the proposed method. Furthermore, it is unclear how the accuracy of pseudo-labels affects the final performance. In detail, in Equation (6), you mix labeled and unlabeled samples with the same class up, whose performance may heavily rely on the quality of the pseudo labels. Additionally, in the t-SNE Visualization, I want to know if Equation (8) contributes to the representation most.
>
> To assess the impact of Prototype Merging Loss (PML) on t-SNE visualization, we included an additional t-SNE visualization in Appendix C of the revised draft. This visualization illustrates the performance of our algorithm without the implementation of PML. In contrast to the distinct clustering observed in Fig. 2, the samples are only loosely clustered around their respective prototypes here, resulting in more ambiguous cluster boundaries than those seen with PML. Moreover, several prototypes remain intermingled, demonstrating a lack of distinct discrimination. These observations underscore the vital role of PML, as formulated in Equation (8). PML enhances the algorithm's ability to learn domain-invariant representations by effectively merging prototypes of the same class from diverse domains, while simultaneously ensuring their separation from prototypes of different classes.
>
> [1] Luojun Lin, Han Xie, Zhishu Sun, Weijie Chen, Wenxi Liu, Yuanlong Yu, and Lei Zhang. Semi-supervised domain generalization with evolving intermediate domain. arXiv preprint arXiv:2111.10221, 2023.

---

> ### Author Response · Authors · 2023-11-23
> **With the hope that our response addresses your concerns**
>
> Dear Reviewer sXNN,
>
> As the discussion period draws to a close, we eagerly await your response. We greatly appreciate your valuable contributions to the review and enhancement of our paper.
>
> We have provided detailed responses to each of your concerns. Please review our responses again and kindly let us know whether they adequately address your concerns and if our explanations are heading in the right direction. Any additional feedback would be highly appreciated.
>
> Kind regards,
>
> Authors of Paper1702

---

### Official Review · Reviewer_8HJ1 · 2023-11-03

**Soundness:** 3 good
**Presentation:** 2 fair
**Contribution:** 2 fair
**Rating:** 6
**Confidence:** 3

**Summary:**

The paper provides a new algorithm (ProUD) for the problem of semi-supervised domain generalization. In this problem setting,
we have K different domains, and we have access to labeled data for 1 domain and unlabeled data for the remaining K-1 domains.
The goal is to solve a learning problem with respect to all domains, by leveraging information from both the labeled data and the unlabeled data. This is an important problem, as labeled data may be unavailable for some source domains (a possible case of data inequality).

The paper proposes a new algorithm to solve this task. The algorithm combines different ideas, some are new, and some are an adaptation from previous work, to provide an accurate solution to this problem. Most of the important techniques involve pseudo-labeling to provide labels to the unlabeled data, and a method called Domain mix that tries to control and manage the contribution for very uncertain pseudo-labeling (very important in the first steps of the algorithm, where the pseudo-labels are not going to be precise).

The authors run extensive different experiments on real datasets, comparing their method to a wide range of baselines in the literature. They find that their methods perform on average better than all the baselines. They also run ablation studies to understand and assess the impact of the most important components of their architecture.

**Strengths:**

The paper addresses a very important problem: studying situations when we have limited access to labeled data is of paramount importance in practice.

From my perspective, the algorithm has two strengths. (1) it combines different techniques in order to provide a good solution to this problem, and it verifies that each of those individual components plays an important role in the ablation study, (2) it provides extensive analysis and comparison with the state-of-the-art to verify the superiority of their method.

Apart from a few details (see Weaknesses and comments below), I think the presentation of the paper is good.

**Weaknesses:**

It is not clear to me why your architecture only learns a single model, and what is the motivation behind it. Although there may be domain-invariant features that we can learn from multiple sources, different features may provide different information depending on the domain.
As an example assume that there are two features f_A and f_B,  domain A may get a very good classification from a feature f_A (and not f_B), and domain B may get a very good classification from a feature f_B (and not f_A). In this case, it would make more sense to build different models that can exploit different features across the different domains. In your algorithm, this means that we keep the feature extractor g equal, but we build a function h for each different domain (similar strategies are also applied in ZSL).

In the data inequality model, some data sources may have a different number of examples. In particular, it could be that we have access to comparatively fewer data points for a given unlabeled domain. It looks to me that your model gives the same weight to each sample from each domain. In this case, if an unlabeled domain t is unrepresented (we have less data  N_t from it), then your model would still suffer from a data inequality issue, as the loss would be less influenced by the fewer samples on this unlabeled domain.


On the experiments:
- Is it the accuracy with respect to a held-out dataset? Or is the model evaluated on the same unlabeled data that is also used during training?
- Why is the accuracy averaged over the last 5 train epochs rather than only on the last epoch?
- I cannot find an explanation on how the hyper-parameters are chosen for your algorithms (is there a validation step?), and how the hyper-parameters are chosen for the baseline algorithms. I think this is important when evaluating methods on a new dataset (without overfitting due to the choice of the hyper-parameters).
- I believe that the standard deviation for the average accuracy is not reported. In particular, you obtain an average accuracy over 3 runs (with 3 different seeds), but it is not clear what is the variance of this value, which is important for comparison with other methods. (The reported STD is across the domain combinations rather than on the 3 seeded runs).

**Questions:**

(1) See the question above on the experiments, in particular for the standard deviation, hyper-parameters, and the choice of how the accuracy is reported.

(2) See the points above on the proposed model/architecture. Why do you learn a single model for each domain, and how do you handle the case when an unlabeled domain is under-represented (it has fewer samples than other unlabeled domains)?

------------



A couple of suggestions:
- "dist is a function to measure the cosine distance" -> "dist is the cosine distance".
- It would be useful for the reader to get some intuitions behind some equations (in simple words), such as (1) and (3).
- I would add a very synthetic explanation of what a "prototype" is for clarity.
- I would briefly clarify that the shift that you are considering is only on the distribution of the features (unless I am missing something), but the classification problem is the same across all tasks.
- There are other settings that are "similar" to this that would maybe be worth discussing in the introduction / related work as further motivation. One setting is Zero-shot Learning (ZSL), where we do not have access to unlabeled data for the other domains (and the classification may change, and one is provided with a description of the classes). Another setting is (programmatic) weak supervision, where the goal is to design simple rules to label an unlabeled domain, and pseudo-labeling and noise-aware losses are also used (e.g., see [A]).

[A]: Ratner, Alexander, et al. "Snorkel: Rapid training data creation with weak supervision." Proceedings of the VLDB Endowment. International Conference on Very Large Data Bases. Vol. 11. No. 3. NIH Public Access, 2017.

---

> ### Author Response · Authors · 2023-11-20
> **Detailed Explanation of the Experiments**
>
> Thank you for your insightful review of our paper. We appreciate the depth of your analysis and your recognition of the significance of our work in addressing the crucial challenge of data inequality. Your constructive feedback on the strengths and weaknesses of our approach, as well as your questions and suggestions, are highly valuable for refining our research. We will carefully address all your questions below.
> >(1) See the question above on the experiments, in particular for the standard deviation, hyper-parameters, and the choice of how the accuracy is reported.
>
> >Is it the accuracy with respect to a held-out dataset? Or is the model evaluated on the same unlabeled data that is also used during training?
>
> We evaluate our model on an unseen test domain, distinct from the unlabeled domains used during training. For instance, in a dataset comprising four domains, we designate one domain as the labeled source domain, two domains as unlabeled source domains, and the remaining domain as the unseen target domain.
>
> >Why is the accuracy averaged over the last 5 train epochs rather than only on the last epoch?
>
> We report the average accuracy over the last five training epochs, following the approach of EID [1], the state-of-the-art baseline for SSDG, to ensure a fair comparison. For our method, the difference between the average accuracy of the last five epochs and the accuracy of the final epoch is negligible.
>
> >I cannot find an explanation on how the hyper-parameters are chosen for your algorithms (is there a validation step?), and how the hyper-parameters are chosen for the baseline algorithms. I think this is important when evaluating methods on a new dataset (without overfitting due to the choice of the hyper-parameters).
>
> We split source domain dataset into training and validation sets, using the validation set for hyper-parameter tuning. The evaluation is conducted on an unseen test domain dataset.
>
> >I believe that the standard deviation for the average accuracy is not reported. In particular, you obtain an average accuracy over 3 runs (with 3 different seeds), but it is not clear what is the variance of this value, which is important for comparison with other methods. (The reported STD is across the domain combinations rather than on the 3 seeded runs).
>
> The standard deviations of the average accuracy over 3 seeds are marginal, and the lower bounds of the average accuracy $(\mu- \sigma)$ still outperform all the baselines. The detailed experimental results are presented below.
> | Seed |  PACS | Office-Home | Digits-DG |
> |:----:|:-----:|:-----------:|:---------:|
> | 2022 | 75.08 |    56.33    |   74.37   |
> | 2023 | 75.09 |    56.24    |   74.26   |
> | 2024 | 75.02 |    56.28    |   74.10   |
> | Avg. | 75.06 |    56.28    |   74.24   |
> | Std. |  0.03 |     0.04    |    0.56   |
>
> [1] Luojun Lin, Han Xie, Zhishu Sun, Weijie Chen, Wenxi Liu, Yuanlong Yu, and Lei Zhang. Semi-supervised domain generalization with evolving intermediate domain. arXiv preprint arXiv:2111.10221, 2023.

---

> ### Author Response · Authors · 2023-11-20
> **Detailed Explanation of the Proposed Model**
>
> >(2) See the points above on the proposed model/architecture. Why do you learn a single model for each domain, and how do you handle the case when an unlabeled domain is under-represented (it has fewer samples than other unlabeled domains)?
>
> >It is not clear to me why your architecture only learns a single model, and what is the motivation behind it. Although there may be domain-invariant features that we can learn from multiple sources, different features may provide different information depending on the domain. As an example assume that there are two features $f_A$ and $f_B$, domain $A$ may get a very good classification from a feature $f_A$ (and not $f_B$), and domain $B$ may get a very good classification from a feature $f_B$ (and not $f_A$). In this case, it would make more sense to build different models that can exploit different features across the different domains. In your algorithm, this means that we keep the feature extractor $g$ equal, but we build a function $h$ for each different domain (similar strategies are also applied in ZSL).
>
> Thank you for your insights. I agree that it is feasible to train a single feature extractor, $g$, along with different classifiers, $h_i$, for distinct source domains. This approach resembles a multi-task learning setting, albeit focused on multi-domains rather than multi-tasks, which may lead to optimal performance for each domain with its corresponding classifier. However, it is important to note that the ultimate goal in SSDG is Domain Generalization, aiming to perform well on “unseen” domains (e.g., after training with domains $A$, $B$, and $C$, we evaluate test performance on domain $D$). If we train multiple classifiers, such as {$h_A$, $h_B$, $h_C$}, we face the issue of deciding which classifier to use for a newly provided domain $D$.
>
> On the other hand, you might suggest utilizing different domain adaptation models for each unlabeled domain to generate pseudo-labels, so we can apply domain generalization to pseudo-labeled datasets. Indeed, this approach has been proposed in the previous work [1] on SSDG. However, it requires separate domain adaptation models and their respective training processes for each unlabeled source domain, to address SSDG. This imposes a significant limitation on its practical application to SSDG problems with a large number of source domains. The same challenge applies to attempts to exploit different combinations of state-of-the-art (SoTA) models for Domain Generalization (DG) and Domain Adaptation (DA). Therefore, our single model-based approach offers a significant practical advantage in handling numerous source domains in a scalable manner.
>
> >In the data inequality model, some data sources may have a different number of examples. In particular, it could be that we have access to comparatively fewer data points for a given unlabeled domain. It looks to me that your model gives the same weight to each sample from each domain. In this case, if an unlabeled domain t is unrepresented (we have less data $N_t$ from it), then your model would still suffer from a data inequality issue, as the loss would be less influenced by the fewer samples on this unlabeled domain.
>
> We appreciate your constructive feedback. This is an important point that has been under-emphasized in our paper, yet it has been successfully addressed with our method. Our DomainMix method not only modulates the involvement of samples from each unlabeled domain based on their uncertainty, but also enhances the diversity of samples from under-represented unlabeled domains by mixing them with labeled samples. A similar approach has proven effective in previous work on few-shot domain adaptation [2], which paired a few target domain samples with a large number of source domain samples to learn domain-invariant features across both domains. Furthermore, our Prototype Merging Loss (PML) constructs  mean prototypes ($\bar{C}_k$) by assigning equal weights to the same class prototypes from each domain, regardless of the number of samples each domain contains. This offers a significant advantage in representing under-represented domains, as it prevents representation bias due to the uneven sample distribution across domains.
>
> [1] Luojun Lin, Han Xie, Zhishu Sun, Weijie Chen, Wenxi Liu, Yuanlong Yu, and Lei Zhang. Semi-supervised domain generalization with evolving intermediate domain. arXiv preprint arXiv:2111.10221, 2023.
>
> [2] Saeid Motiian, Quinn Jones, Seyed Iranmanesh, and  Gianfranco Doretto. Few-shot adversarial domain adaptation. Advances in neural information processing systems 30, 2017.

---

> ### Author Response · Authors · 2023-11-20
> **A Response to Suggestions**
>
> >"dist is a function to measure the cosine distance" -> "dist is the cosine distance".
>
> >It would be useful for the reader to get some intuitions behind some equations (in simple words), such as (1) and (3).
>
> >I would add a very synthetic explanation of what a "prototype" is for clarity.
>
> We sincerely appreciate your thorough and insightful suggestions for our work. We have uploaded the revised version of the draft. Modifications have been made based on your suggestions, which can be found in 'Prototype-based Pseudo-labeling' of Section 3.2.
>
> >I would briefly clarify that the shift that you are considering is only on the distribution of the features (unless I am missing something), but the classification problem is the same across all tasks.
>
> The distribution shift we consider is on image space (input level), and the classification problem is same across all domains.
>
> >There are other settings that are "similar" to this that would maybe be worth discussing in the introduction / related work as further motivation. One setting is Zero-shot Learning (ZSL), where we do not have access to unlabeled data for the other domains (and the classification may change, and one is provided with a description of the classes). Another setting is (programmatic) weak supervision, where the goal is to design simple rules to label an unlabeled domain, and pseudo-labeling and noise-aware losses are also used (e.g., see [A]).
>
> Thank you for your valuable insights deepening our discussion on the relevant studies. We will need some time to familiarize ourselves with the recent developments in the areas you suggested. These discussions will be included in the 'Related Work' section of our final paper. If you have any additional suggestions or references, please share them with us. Your input would be greatly appreciated and would contribute to the enrichment of our paper.

---

> ### Author Response · Authors · 2023-11-23
> **With the hope that our response addresses your concerns**
>
> Dear Reviewer 8HJ1,
>
> As the discussion period draws to a close, we eagerly await your response. We greatly appreciate your valuable contributions to the review and enhancement of our paper.
>
> We have provided detailed responses to each of your concerns. Please review our responses again and kindly let us know whether they adequately address your concerns and if our explanations are heading in the right direction. Any additional feedback would be highly appreciated.
>
> Kind regards,
>
> Authors of Paper1702

---

### Meta-Review · Area_Chair_nfwD · 2023-12-06

**Metareview:**

This paper addresses the problem of semi-supervised domain generalization by studying the data inequality problem across domains. A novel algorithm is proposed and evaluated on different benchmarks. The method is based on the use of domain-aware prototypes and uncertainty strategies.

On the positive side, the reviews have mentioned that the paper studies an important and interesting problem for domain generalization where only one domain is labeled, it provides a good solution thanks to a sound combination of techniques, the paper is well presented/written.
On the negative side, the proposed solution is not justified enough, the method is incremental by combining existing methods, the contribution lacks of theoretical justification, the experimental evaluation lacks of some existing baselines and known datasets/benchmarks and improvements seem relatively limited.

The authors provided a rebuttal by providing answers to the remarks raised by the reviewers. They have provided additional explanation on the contribution and the methodology and gave complementary experimental results.
During the discussion, the interestingness of the topic addressed by the paper has been unanimously identified. However, reviewers also agreed on the lack of clear innovation of the methodology proposed which combines different existing approaches. This combination may be not trivial, but the paper does not provide enough clear arguments for supporting the innovative aspect of the approach. There is not clear insights for future work aiming at combining methods for DA and DG. The theoretical analysis can be made more precise and insightful and the experiments could benefit from additional evaluation on DomainNet for example.

Overall, the contribution of the paper does not appear to be sufficient for this ICLR venue, I propose then rejection.
Nevertheless, I encourage the authors to improve their methodology for other venues, the problem studied the paper is indeed very interesting.

**Justification For Why Not Higher Score:**

There was a clear consensus among the reviewers that the contribution lacks of clear justification of the novelty of the contribution which appears as a combination of existing techniques.

**Justification For Why Not Lower Score:**

N/A

---

### Decision · Program_Chairs · 2024-01-16

Reject